# GAMBIT (Genomic Approximation Method for Bacterial Identification and Tracking): A methodology to rapidly leverage whole genome sequencing of bacterial isolates for clinical identification

**Jared Lumpe**[1]*, **Lynette Gumbleton**[2], **Andrew Gorzalski**[2], **Kevin Libuit**[3], **Vici Varghese**[4], **Tyler Lloyd**[4], **Farid Tadros**[5], **Tyler Arsimendi**[5], **Eileen Wagner**[3], **Craig Stephens**[5], **Joel Sevinsky**[3], **David Hess**[2,5,6]*, **Mark Pandori**[2,4,6,7]*

1 Independent Researcher, Meriden, Connecticut, United States of America, 2 Nevada State Public Health Laboratory, Reno, NV, United States of America, 3 Theiagen Consulting LLC, Highlands Ranch, CO, United States of America, 4 Alameda County Department of Public Health, Oakland, CA, United States of America, 5 Biology Department, Santa Clara University, Santa Clara, CA, United States of America, 6 Department of Pathology and Laboratory Medicine, University of Nevada, Reno School of Medicine, Reno, NV, United States of America, 7 Department of Microbiology and Immunology, University of Nevada, Reno School of Medicine, Reno, NV, United States of America

* jared@jaredlumpe.com (JL); mpandori@med.unr.edu (MP); dhess@unr.edu (DH)

## Abstract

Whole genome sequencing (WGS) of clinical bacterial isolates has the potential to transform the fields of diagnostics and public health. To realize this potential, bioinformatic software that reports identification results needs to be developed that meets the quality standards of a diagnostic test. We developed GAMBIT (Genomic Approximation Method for Bacterial Identification and Tracking) using k-mer based strategies for identification of bacteria based on WGS reads. GAMBIT incorporates this algorithm with a highly curated searchable database of 48,224 genomes. Herein, we describe validation of the scoring methodology, parameter robustness, establishment of confidence thresholds and the curation of the reference database. We assessed GAMBIT by way of validation studies when it was deployed as a laboratory-developed test in two public health laboratories. This method greatly reduces or eliminates false identifications which are often detrimental in a clinical setting.

## Introduction

Bacterial species that cause human disease are an ever evolving threat and a leading cause of death world-wide [1]. Because of this evolving threat, identification and characterization of these pathogens are crucial for effective infection control methods [2,3]. The potential of whole-genome sequencing (WGS) of clinical isolates to rapidly identify bacterial species, detect antibiotic resistance and provide relatedness information for epidemiological purposes is broadly recognized in the field of clinical microbiology [4–7]. Molecular diagnostic tools

**Data Availability Statement:** Data are available in the Supporting Information files and from NCBI

(Bioprojects: PRJNA857686, PRJNA218110, PRJNA230403, PRJNA239251, PRJNA266293, PRJNA212117, PRJNA290730 and PRJNA211456).

**Funding:** This work was supported in part by the National Institutes of Health, grant R15AI130816-01A1. Funds provided by the Department of Homeland Security in the form of Urban Area Security Initiative (UASI) funding also supported this study. The funders had no role in study design, data collection and analysis, decision to publish, or preparation of the manuscript.

**Competing interests:** The authors have declared that no competing interests exist.

**Abbreviations:** WGS, Whole-genome sequencing; ANI, Average Nucleotide Identity.

based on nucleic acid identification have long been used in clinical microbiology. Clinical microbiology represents approximately 70% of the global market for molecular diagnostic tools, so the extension of WGS as a diagnostic tool is a natural fit [8,9]. WGS has been deployed in clinical microbiology laboratories as proof-of-concept, used for surveillance of antibiotic resistance or focused on a specific genus or species of clinical relevance, but the literature does not detail the routine use of WGS by clinical microbiology laboratories for identification of bacterial isolates [7,10,11].

Over the past decade, improvements to Next Generation Sequencing (NGS) have dramatically increased access to draft genome sequences (especially in bacteria) and have substantially decreased in cost per genome [10,12]. This technology has led to expanded use of whole genome sequencing (WGS) in microbial research [13]. This research has produced expanded insights into microbial diversity [14,15], antibiotic resistance [16–18], the human microbiome and the evolution and spread of microbial pathogens [19–22].

Several problems have been noted in the literature that prevent routine use of WGS for clinical microbiology. Consistent among reviews on the topic is that bioinformatic pipelines are needed that process the genomic data once generated [10,11,23]. As Besser [10] notes, "It is a challenge that the comparability of the sequence data generated on different platforms with different error profiles using different library preparation methods has still not been comprehensively assessed and validated." The method described herein is agnostic to the platform on which the raw sequence data was generated, which allows the use of bacterial genomes generated from a variety of whole genome sequencing methods.

In this manuscript we describe a methodology to rapidly compare a single generated bacterial genome sequence to a reference database of 48,224 bacterial isolates spanning 1,415 species. We describe three advancements that in combination were required to build upon existing k-mer based strategies to allow GAMBIT to possess the quality control parameters desired for its use as a diagnostic laboratory-developed test. Firstly, a compressed k-mer based format enabled us to store over 48,000 bacterial genomes from the NCBI RefSeq database—a majority of the genomes available at the time of development. Secondly, significant curation was required to remove ambiguous or potentially incorrectly labeled genomes, which greatly increased confidence in positive matches. Lastly, we used this curated version of the NCBI RefSeq database and our scoring method to generate confidence thresholds for identification of each individual species and genus, accounting for the large variations in diversity between taxa. Thus, the end-user does not rely simply on the closest match, but is informed whether that closest match exceeds a threshold for highly confident identification. Furthermore, we detail the deployment of this methodology in two clinical settings: The Alameda County Public Health Laboratory and the Nevada State Public Health Laboratory. The results presented here demonstrate that our methodology provides the necessary bioinformatic elements necessary for use of WGS in a clinical microbiology laboratory.

## Results

### A k-mer-based representation supports a reference library of over 48,000 bacterial genomes

Whole-genome sequencing (WGS) data have been used for species identification using k-mer based methods [21,24,25]. A previously described k-mer based strategy of identification [24] was limited to ~1,700 reference genomes based on their data storage structures. GAMBIT was able to extend its reference database to 48,224 bacterial genomes by utilizing a different method of representing genomic data. GAMBIT finds all 11-mers in a genome assembly that immediately follow the prefix sequence ATGAC. There are only $4^{11}$ = 4,194,304 such

sequences. By associating each of these 11-mers with an integer in the range $1..4^{11}$ corresponding to its position in alphabetical order, the set of unique 11-mers present in a genome can be represented with a simple integer array. This storage method allows the entire database of 48,000+ reference genomes to fit in less than 1.4GB, and supports rapid screening of unknown genomes against the full set of references. The entire GAMBIT classification process took an average of less than .5 seconds per genome when run on a personal laptop for data sets 1–4, suggesting that the process can be scaled to far larger reference databases.

## GAMBIT genomic distance metric correlates with sequence identity

We utilized four sets of bacterial genome assemblies in validating the GAMBIT distance metric (Table 1). The first two consist of high-quality assemblies obtained from the NCBI RefSeq database, while the second two are derived from clinical samples. Set 1 (S1 Table) consists of 492 *E. coli* genomes used in Ondov *et al* [25] to validate the Mash tool, which defines a similar k-mer based genomic distance metric. Set 2 (S2 Table) is composed of 70 completely closed genomes that broadly cover the bacterial kingdom based on Konstantinidis and Tiedje [26]. Bacterial groups include 15 genomes from *Enterics*, 9 from *Streptococcus*, 7 from *Staphylococcus*, 4 from *Bacillus*, 4 from *Mycobacterium*, 4 from *Neisseria*, 3 from *Bordetella*, 3 from *Pseudomonas* and 21 other genomes representing the *following g*enera: *Brucella*, *Burkholderia*, *Clostridium*, *Heliobacter*, *Legionella*, *Rickettsia*, *Tropheryma*, *Vibrio*, *Xanthomonas* and *Xylella*. Set 3 (S3 Table) consists of 88 "gold standard" proficiency test samples, described in detail later in the manuscript. The source of these proficiency test samples is described in the Materials and Methods. They are part of the routine testing for clinical diagnostic laboratories spanning 46 different species over 28 unique genera. Lastly, Set 4 (S4 Table) is a set of 604 genomes obtained between 2020–2022 by the Nevada State Public Health Laboratory. It contains 25 species and 15 genera.

We used ANI (Average Nucleotide Identity) as a baseline measure of genomic similarity to validate the GAMBIT distance metric. ANI is generally used to determine similarity at the species or genus level with thresholds above 0.92 being optimal for species-level calls [28]. We compared ANI values against GAMBIT distances for all pairs of genomes in each of our data sets 1–4 (Fig 1). Spearman correlation was high in all four data sets (Set 1 = -0.977; Set 2 = -0.968; Set 3 = -0.969; Set 4 = -0.979) for comparisons in which the ANI was reported by the FastANI tool (100%, 5.59%, 7.42% and 47.4%). Of note for Set 4, GAMBIT demonstrates broader descrimination for genome pairs with ANI near 100%, resulting in a number of values off the major trendline. Additionally, we show that GAMBIT recovers distances between highly dissimilar genome pairs that distribute over a wide range in instances that the FastANI tool does not return a value because the genomes are two dissimilar (S2 Fig). FastANI's cutoff of approximiately 80% corresponds to a GAMBIT distance of approximately 0.99. This demonstrates that GAMBIT calculates relatedness over a much broader taxonomic range than FastANI.

**Table 1. List of genome sets used in this manuscript.**

| Data Set | Number of Genomes | Source of fasta/fastq files | Phylogenetic Diversity | Assembly Quality | Reference |
|---|---|---|---|---|---|
| Set 1 | 492 | NCBI Ref Seq | Low (*E. coli* only) | Medium | [25] |
| Set 2 | 70 | NCBI Ref Seq | High (multiple phyla) | High (all reference genomes) | [26] |
| Set 3 | 88 | Alameda County Public Health Labs | High (multiple phyla) | Medium | none |
| Set 4 | 604 | Nevada State Public Health Labs | High (multiple phyla) | Medium | none |
| Set 5 | 29 | Stephens Lab | Low (*E. coli* only) | High (all closed genomes) | [27] |

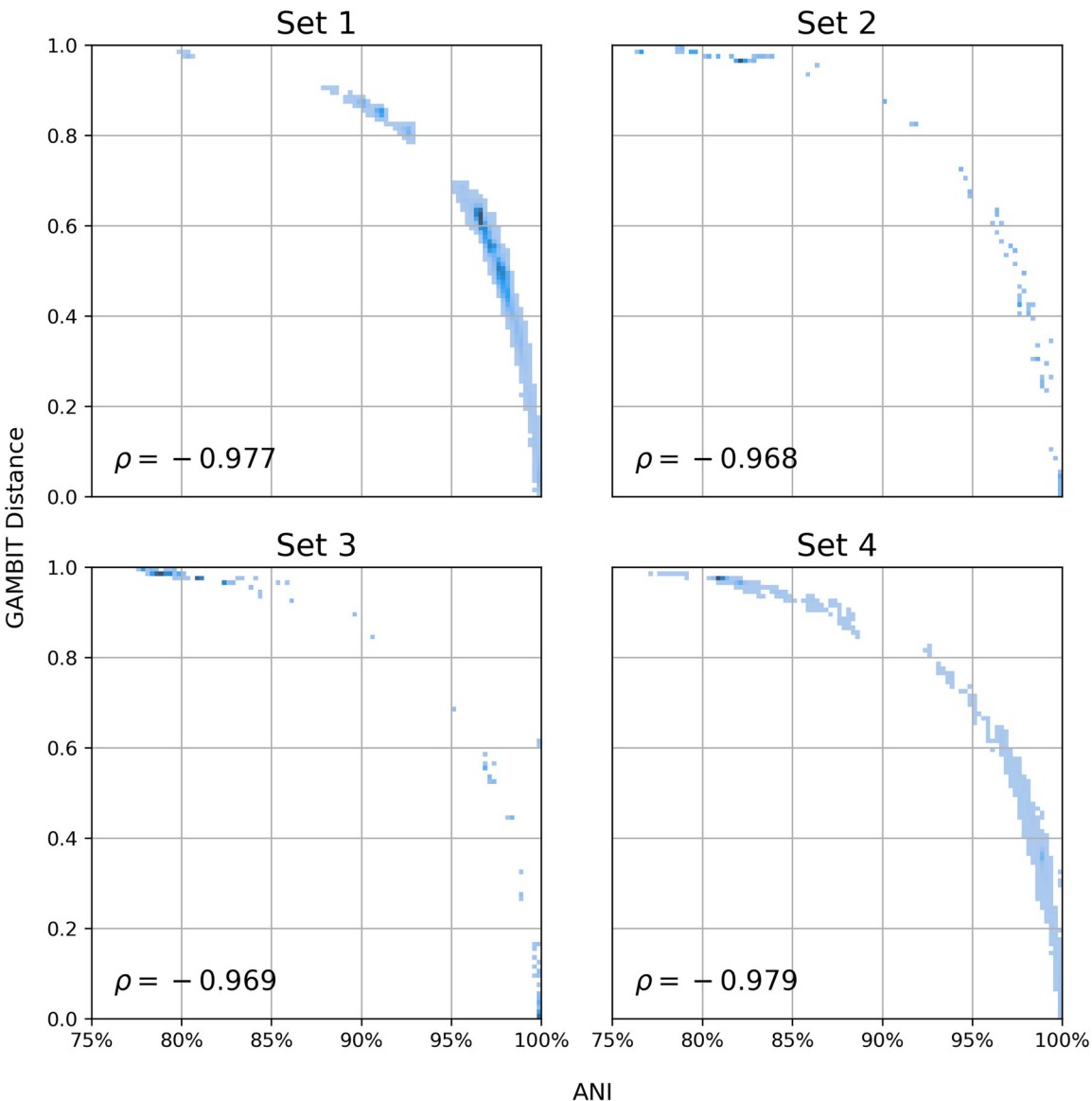

**Fig 1. Relationship between GAMBIT distance and ANI (Average Nucleotide Identity) for all pairs of genomes in data sets 1–4.**
The relationship is nonlinear but very close to monotonic as measured by Spearman correlation (shown in the bottom left corner of each subplot). ANI was calculated using the FastANI tool (see Materials and methods) with default parameter values. GAMBIT distances were calculated using the same parameter values as in the RefSeq reference database (k = 11, prefix = ATGAC). As FastANI only reports ANI values greater than ~80%, the fraction of total pairwise comparisons shown here were 100%, 5.5%, 7.4% and 47.4% for data sets 1–4 respectively. GAMBIT distances for comparisons with no ANI value to compare to are shown in S2 Fig.

## GAMBIT distances are robust with regard to parameter choice

We sought to evaluate the dependence of the GAMBIT distance metric on its two parameters: the prefix sequence and the length of the k-mers used. In our reference database and elsewhere in this paper we use the prefix ATGAC and a k value of 11 based on work from the Aarestrup group [24]. We added two random nucleotides to this default prefix and generated an additional seven fully random 7bp prefix sequences, and tested them truncated to lengths of 4, 5, 6 and 7. In combination with variations in the prefix sequence we also tested odd values of k from 7 to 17. We evaluated the performance of each set of parameter values as in the previous

section, by calculating all pairwise GAMBIT distances for each of our test genome sets and determining the Spearman correlation with ANI. In Fig 2 we show results for a limited set of prefix values along with all values of k. Fig 2A displays variations of the default prefix to different lengths, and Fig 2B compares the default prefix to random sequences of the same length. The full combined set of prefix variations are shown in S1 Fig. All panels in these figures use the same axes and are comparable.

The set of parameters used in the final version of GAMBIT are shown in Fig 2A on the plot with the blue highlighted border. Beginning with this plot, we would expect to see the highest Spearman correlation in the upper right corner of the plot--which corresponds to the longest length of those k-mers (the longer the k-mers, the more of the genomic information is being retrieved). These data points represent the maximum amount of genomic information being used by the GAMBIT algorithm and they do represent the highest Spearman correlations for each set of variables. In terms of k-mer length, performance increases in all instances as k-mer length increases reaching a plateau at a k-mer length of 11. This suggests that our chosen k-mer length of 11 balances discriminatory power with speed of the algorithm.

Comparing all four panels in Fig 2A, we observe the effect of changing the length of the prefix. We predict that as prefix length increases correlation should decrease because less of the genome is being sampled. Interestingly, correlations do not decrease significantly when increasing the prefix length to 5 (the chosen parameter for our version of GAMBIT), but degradation in performance is seen when the prefix length is increased to 6 or 7 nucleotides.

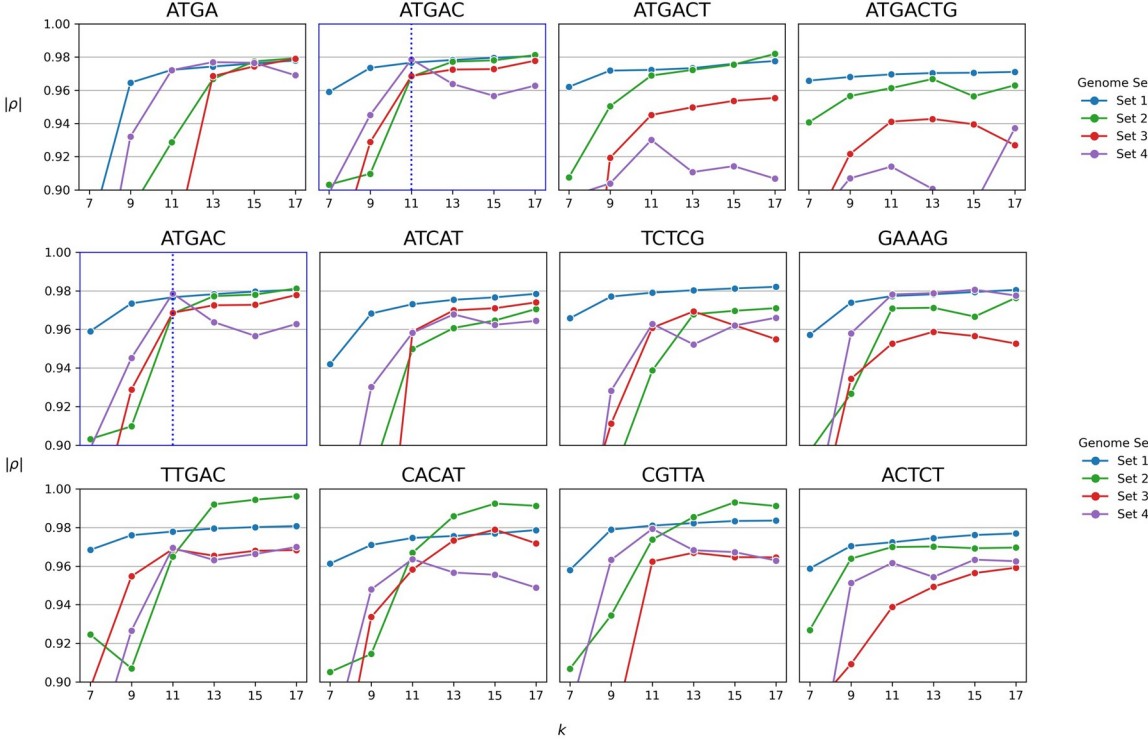

**Fig 2. Spearman correlation between GAMBIT distance and ANI for different GAMBIT parameter values.** Each subplot in both A and B represents a different choice of prefix sequence. Subplots show the absolute value of the Spearman correlation ρ vs. value of the k parameter for all pairwise comparisons within genome sets 1–4 (Table 1). Standard values of the prefix and k parameters used throughout the rest of the manuscript are highlighted by a blue subplot border and blue vertical line, respectively. A. Variations of our standard prefix ATGAC with length between 4 and 7 nucleotides. B. Standard prefix sequence plus 7 random sequences of the same length. See S1 Fig for the full set of variations in prefix sequence and length.

Lastly, we observe the effect of the actual sequence of the prefix sequence. We compare our prefix sequence (at length 5) to seven other randomly generated prefix sequences (Fig 2B). We compared Spearman correlation between ANI and GAMBIT distance for each sequence over k-mer lengths ranging from 7 to 17. Our chosen prefix sequence (ATGAC), performs in the middle of the different prefix sequences across all comparison sets. Without understanding why specific prefix sequences perform better or worse, using a sequence that is consistently in the middle should minimize the chance of introducing bias into our method based on sequences overrepresented in our comparison dataset but not present broadly among all bacterial species. A more exhaustive comparison of parameters is shown in S1 Fig.

## Curated, high-quality database construction based on RefSeq

There is no large-scale, gold-standard reference database of bacterial genomes. The NCBI RefSeq database contains a large and wide array of genomes, but is not curated for accuracy. Our goal in creating a database for GAMBIT was two-fold: first, that the genomes representing each species in our database must contain enough distinction from other species that similarity thresholds could be extracted (see next section); second, that we have the highest level of confidence that a genome in our database was actually from the species that matched its labeled identification. Interrogation of the RefSeq database clearly identified incidents of mislabeled genomes. The conservative set of filters used to achieve our two stated goals are described below. To construct our reference database, we started with the 60,857 available bacterial genomes from the NCBI RefSeq database [29] on July 1st, 2016. We purged the 3,996 genomes that did not have an associated genus and/or species. Additionally, we purged the 3,305 genomes that were the only representative of its species, as we required at least two separate sequenced isolates in order to determine a classification threshold for each species. After the removals noted above, we calculated all pairwise GAMBIT Distances for the remaining 53,556 genomes. We then began an iterative process of removing ambiguous genomes, resulting in the removal of an additional 5,332 genomes. We defined ambiguous genomes as genomes that met any of the three following criteria: (1) any genome that did not cluster well with the majority of the other genomes within their species, (2) any genome that clustered well with some members of their species but also several members of another species in the database, or (3) any genome that did not cluster well with any genomes in the database. We chose to eliminate these genomes to attain a higher level of certainty in the final database, which contains 48,224 bacterial genomes representing 1,414 species and covering 454 unique genera. The remaining genomes in our reference database represent a large set of genomes over a large breadth of clinically relevant species (genome and taxon lists in S5 and S6 Tables). This gold-standard of bacterial genomes and their genus and species identification should be a valuable resource for other microbial genomic projects.

## The necessity of draft genome assembly for generated FASTQ files

Analysis of the copy numbers of unique k-mers found in raw sequencing data (Fig 3) reveals a bimodal distribution. This is shown for five different whole-genome sequences selected from Set 3. Histograms are split based on whether k-mers were present (green) or not present (blue) in the assembled genome sequence. The high copy number peak consists almost entirely of k-mers retained in the assembly and is centered around the sequence coverage depth (represented by the dashed black vertical lines). This is expected as sequence coverage depth represents the average number of sequence reads that cover a particular sequence, thus the number of times each unique k-mer is identified in a given sequencing experiment should be approximately equal to the sequence coverage depth.

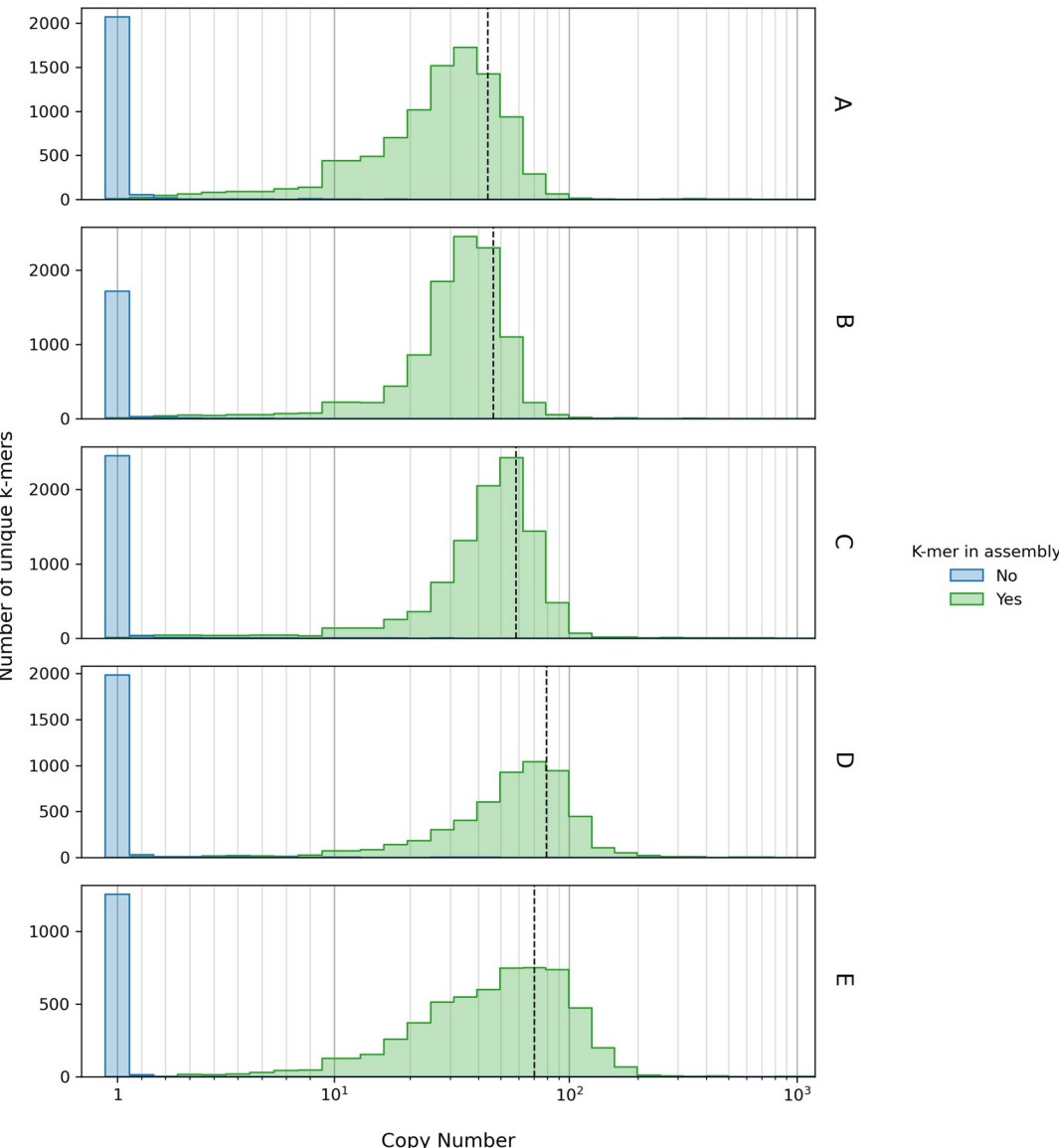

**Fig 3. Distribution of k-mer counts in raw sequencing reads.** Raw FASTQ files for a selection of Set 3 genomes were scanned for k-mer matches using the standard GAMBIT procedure. After discarding matches containing nucleotides with PHRED scores less than 20 the number of copies of each unique k-mer found was tallied. Species were *P. stuartii*, *E. coli*, *Y. enterocoliticia*, *E. faecalis* and *S. epidermis* for subplots A-E, respectively. The blue histograms show the copy number of k-mers found in the raw sequencing reads that are not recovered after assembly, and likely represent sequencing errors. The green histograms show the copy number of k-mers found in the sequencing reads that are present in the final genome assembly. The center of these distributions closely tracks the sequencing depth (denoted by the black dashed vertical line). Note that histogram bins encompass single integers from 1–10 and are thereafter logrithmically spaced.

The low copy number peak consists almost entirely of k-mers which were predominantly found only once in the FASTQ file and were excluded from the final assembly, with a short right-sided tail. This distribution likely predominantly represents sequencing errors, which would account for copy numbers far below the coverage depth and the fact that these k-mers are eliminated after assembly. This is because in generating the consensus sequence in *de novo* genome assembly nearly all sequencing errors are eliminated. For GAMBIT to work properly, users must perform an assembly of the FASTQ file to eliminate sequencing errors by consensus.

## Establishing confidence thresholds for classification

To utilize GAMBIT as a diagnostic test, it was crucial to establish thresholds for definitively calling a particular genome at the species or genus level. As described in the section titled "GAMBIT taxonomic classification", GAMBIT classifies unknown genomes by finding the distance to the closest reference genome and comparing that distance against the thresholds of the reference genome's species and genus. For most species in our reference database, we were able to use the simplest method which involved calculating two parameters: the maximum intra-species distance ("max intra," or diameter) and minimum inter-species distance ("min inter") (Fig 4). This involves generating GAMBIT distances between each genome in the given species and all other genomes in the reference database. Max intra represents the GAMBIT distance between the two most dissimilar isolates within the species, while min inter represents the closest similarity that can be generated between an isolate in the species of interest and any isolate outside of it. Put another way, min inter represents the best false positive value that could be generated by the 48,224 genomes in our reference database.

For all species we required that the classification threshold be capped at the min inter distance minus a 5% safety margin. This ensures that no isolate in our reference database could generate a false positive using our classification method. Given this limit we sought to set the threshold to the max intra distance of each taxon where possible. This ensured that the threshold of each

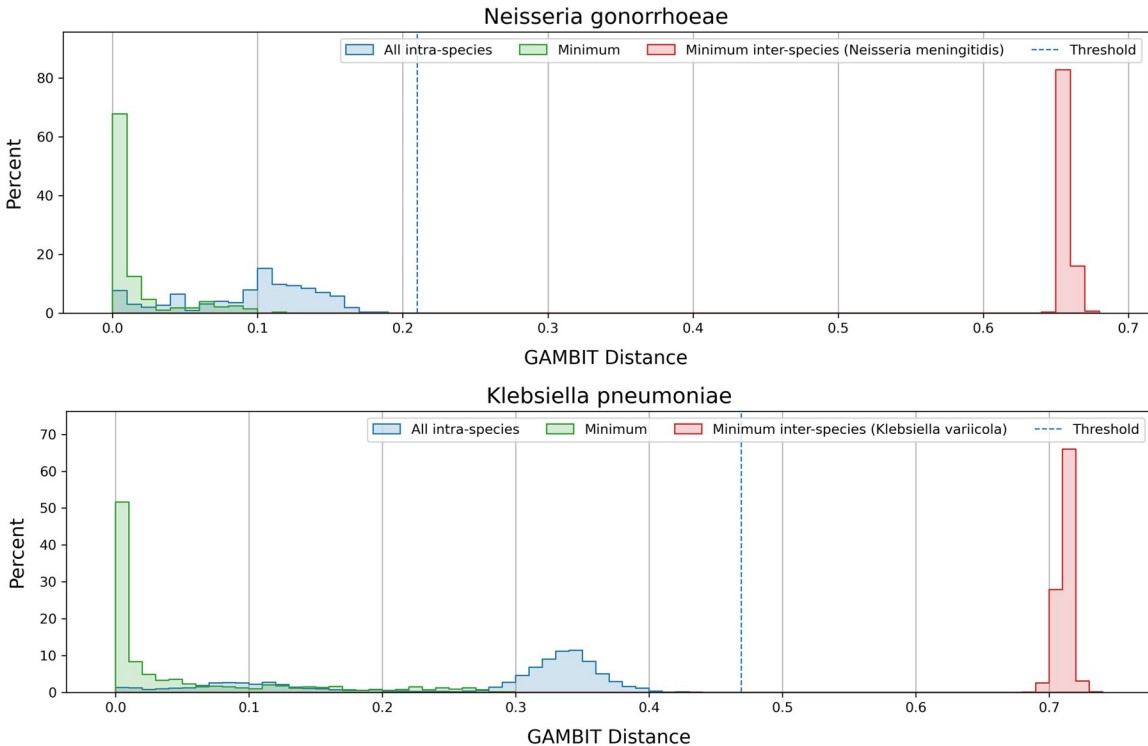

**Fig 4. Distribution of GAMBIT distances within a species and to the nearest sister taxon in the GAMBIT reference database.** Three histograms are shown in each panel (each normalized independently). The green histogram represents the distribution of GAMBIT distances from each reference genome in the species to the closest genome also within the same species. The blue histogram represents the distribution of GAMBIT distances for all pairwise comparisons within the species. The red histogram represents the distribution of GAMBIT distances from each genome in the species of interest to the closest genome in the species' closest sister taxon. The dashed blue line represents the classification threshold for that species in the GAMBIT database, which in both cases was derived from the maximum intra-species distance. Panel A shows Klebsiella pneumoniae and its closest sister taxon Klebsiella variicola, panel B shows Neisseria gonorrhoeae and its closest sister taxon Neisseria meningitidis).

species reflects its level of diversity, which varies significantly from species to species. We chose to be conservative and not exceed the max intra value in order to reduce the possibility of false positives when attempting to classify isolates of species which are not present in our database, a problem which the first rule alone does not address. This goal was possible for most species after performing curation of the reference database genomes as described above.

The GAMBIT distances used to generate thresholds for *Klebsiella pneumoniae* and *Neisseria gonorrhoeae* are plotted in Fig 4 as examples. There are 877 *Klebsiella pneumoniae* isolates in the GAMBIT database. In Fig 4A we plot three distributions of distances. The green distribution represents the top hit (minimum GAMBIT distance) for each of the 877 *Klebsiella pneumoniae* isolates in the database. This is relevant as GAMBIT makes classifications based on the top hit in the database. In all cases the top hit was another isolate within Klebsiella pneumoniae, as expected. The blue distribution shows all pairwise comparisons between the 877 *Klebsiella pneumoniae* isolates (384,126 unique pairs), which yield a max intra value of 0.469. The red distribution shows all pairwise scores between the 877 *Klebsiella pneumoniae* isolates and isolates in its closest sister taxon *Klebsiella variicola*, yielding a min inter value of 0.689. This figure illustrates how most species in our database are well separated from their closest sister taxon. These spreads give us confidence that our thresholds will not generate false positives. Fig 4B shows the same distributions for *Neisseria gonorrhoeae*. There are 208 isolates of *this species* in the GAMBIT database. The max intra (and threshold) for *Neisseria gonorrhoeae* is 0.210 and the min inter to its closest sister taxon (*Neisseria meningitidis*) is 0.649. Of the 1,414 species in our database, we were able to use this method to set the threshold for 1,371 of them (~97%).

Next, we examined 25 species in which the distributions of inter- and intra-species distances overlap to a small degree. In these cases we used 95% of the min inter distance as the threshold as described above. All of the top hits (used to assign species identification by GAMBIT) for genomes in these taxa are well below the resulting thresholds.

Lastly, there were 17 species that had more substantial overlap in their intra and inter distances with one or more sister species (Table 2). For these species we employed the strategy of dividing these species into between 2 and 5 subspecies groups based on clustering of their intra-species distances. We added these subgroups to the database as child taxa to their parent species. These are used for classification using the same method as species and genus taxa, but GAMBIT will report matches to these subgroups as their parent species. At the subspecies level we were able to create thresholds for each subgroup based on either max intra or 95% of min inter (S6 Table). An example of *Shigella boydii* is shown in Fig 5A. *Shigella boydii* had overlaps with *Shigella dysenteriae* shown in the top panel. The distribution of intra distances is shown in blue, while the distribution of inter distances with the overlapping species (in this case *S. dysenteriae*) is outlined in red. However, after dividing *S. boydii* into two subgroups (middle and bottom panels, there is no overlap and clean thresholds can be established. In these panels, distances within a subgroup are shown in blue while distances to the other subgroup are shown in green. The threshold for each subgroup is shown with the vertical dashed blue line. Fig 5B demonstrates how we resolved the overlaps of *Pseudomonas amygdali* with *Pseudomonas savastanoi* (intra distances shown in purple) and *Pseudomonas syringae* (shown in red). Here we divided *Pseudomonas amygdali* into three subgroups, of which subgroups 1 and 3 had no remaining overlaps and subgroup 2 used the 95% of min inter distance to set the threshold.

Two final irregularities had to be addressed during database construction at the species level. First, we could not cleanly differentiate *Lacticaseibacillus casei* and *Lacticaseibacillus paracasei*. Thus, GAMBIT reports the result '*Lacticaseibacillus casei/paracasei*' for top hits that map to the genomes for these two species. Second, while GAMBIT can differentiate between the closely related organisms of *Escherichia coli* and the *Shigella* species, when the top hit for

**Table 2. Species in the GAMBIT database that were broken into subspecies during the process of establishing thresholds.**

| name | max intra | min inter | closest taxon | number of subgroups |
|------|-----------|-----------|---------------|---------------------|
| *Bacillus amyloliquefaciens* | 0.4564 | 0.3473 | *Bacillus velezensis* | 2 |
| *Bacillus cereus* | 0.9020 | 0.5682 | *Bacillus thuringiensis* | 4 |
| *Bacillus pumilus* | 0.8785 | 0.7945 | *Bacillus safensis* | 2 |
| *Brucella suis* | 0.0677 | 0.0220 | *Brucella canis* | 2 |
| *Clostridium botulinum* | 0.9943 | 0.8220 | *Clostridium sporogenes* | 3 |
| *Enterobacter cloacae* | 0.6653 | 0.2893 | *Enterobacter hormaechei* | 4 |
| *Enterobacter hormaechei* | 0.6071 | 0.2893 | *Enterobacter cloacae* | 2 |
| *Escherichia coli* | 0.6825 | 0.3648 | *Shigella sonnei* | 3 |
| *Prochlorococcus marinus* | 0.9989 | 0.9940 | *Arcobacter butzleri* | 4 |
| *Pseudomonas amygdali* | 0.5224 | 0.2951 | *Pseudomonas syringae* | 3 |
| *Pseudomonas putida* | 0.8867 | 0.8334 | *Pseudomonas monteilii* | 3 |
| *Pseudomonas savastanoi* | 0.4898 | 0.2978 | *Pseudomonas amygdali* | 2 |
| *Pseudomonas syringae* | 0.9149 | 0.2951 | *Pseudomonas amygdali* | 5 |
| *Salinispora pacifica* | 0.8048 | 0.5266 | *Salinispora oceanensis* | 3 |
| *Shigella boydii* | 0.2969 | 0.2048 | *Shigella dysenteriae* | 2 |
| *Shigella dysenteriae* | 0.5602 | 0.2048 | *Shigella boydii* | 2 |
| *Streptococcus pseudopneumoniae* | 0.8465 | 0.5806 | *Streptococcus mitis* | 2 |

either of these does not meet the confidence threshold, GAMBIT reports back '*Escherichia coli/Shigella*' rather than *Escherichia* or *Shigella* alone.

If the top hit distance for a new query in GAMBIT is larger than the species threshold, there is still a possibility that GAMBIT can make a call at the genus level. We repeated the same procedure to set thresholds for all genera. 335 genera had no overlaps between their intra and inter distances,

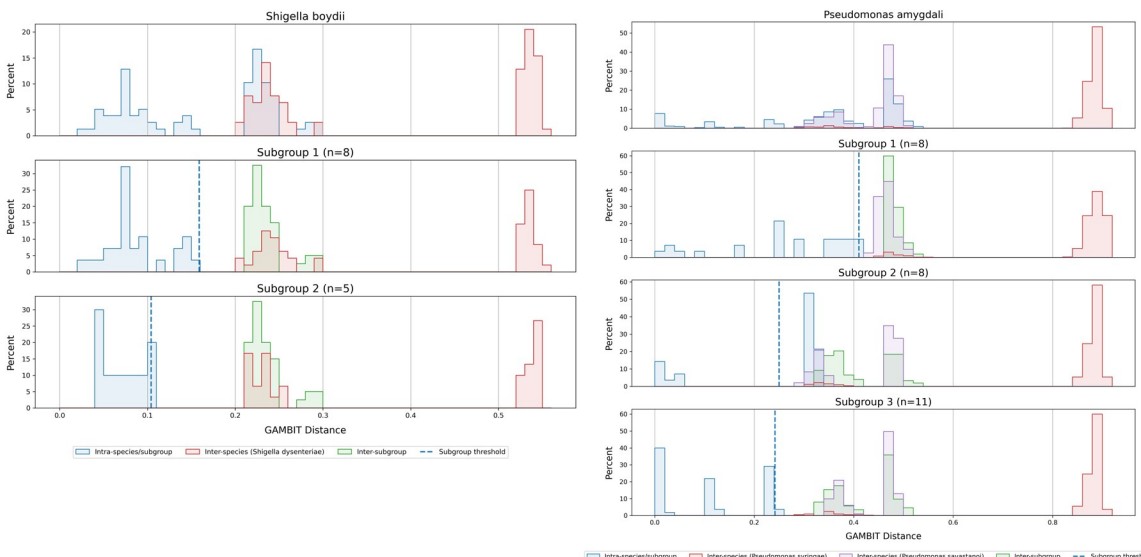

**Fig 5. Distribution of intra- and inter-species GAMBIT distances for two species divided into subgroups in the GAMBIT reference database.** In both subplots, the top panel shows the distribution of intra distances in blue and the distribution of inter distances to sister taxa in red and/or purple. The remaining panels show distance distributions for each subgroup. Intra-subgroup distances are shown in blue, inter-subgroup distances in green, and inter-species distances are as in the top panel. Subgroup thresholds are represented with a dashed blue line. A. Shigella boydii. Inter-species distances with Shigella dysenteriae are shown in red. B. Pseudomonas amygdali. Inter-species distances with Pseudomonas savastanoi and Pseudomonas syringae are shown in purple and red, respectively.

while 117 (~26%) had overlaps and used 95% of min inter to set the thresholds. The total number of genera reported is two less than present in the database (452 vs 454) for two reasons. The first is the combining of 'Escherichia and Shigella into a single reporting group as mentioned above. The second is the species *Eubacterium siraeum*. *Eubacterium* is a proposed genus not officially accepted in the NCBI taxonomy, so we only call *Eubacterium siraeum* at the species level.

## Validation of GAMBIT by the Alameda County Public Health Laboratory

From 10/12/16 to 02/13/2020 the Alameda County Public Health Laboratory used single-end whole genome sequencing utilizing GAMBIT for genus and species identification on 88 proficiency test samples (data set 3, See S3 Table). The source of these samples was either isolates identified at the Public Health Laboratory and confirmed by a reference laboratory or specimens provided as quality assessment standards through proficiency testing (College of American Pathologists). Using the conservative thresholds described in the previous sections, GAMBIT reported zero false positives out of these 88 samples (Tables 3 and S7). GAMBIT predicted the correct genus or species needed to pass the proficiency test in 86 of the 88 samples. The remaining 2 samples for which GAMBIT did not make a prediction were correctly predicted and reported using an in silco 16S test on the whole genome sequencing (see Materials and methods for details). In silco 16S was used on an additional sample to report the species level information but only genus level information was expected for the proficiency test (GAMBIT reported Pseudomonas genus, 16S reported Pseudomonas putida). For one of these samples (Proteus vulgaris) the most similar reference genome was of the correct genus but the distance did not meet the threshold. The only case where the top genome match was not of the correct genus was for Granulicatella adiacens. Neither this species nor any member of the genus Granulicatella are in the GAMBIT database and thus no correct match can be expected. In summary, in 87 out of 87 proficiency test samples where the species was present in the GAMBIT database the top match from GAMBIT provided the correct reporting information to satisfy the test. In the only instances the score associated with the top match did not exceed the threshold used by GAMBIT, the top match was correctly confirmed with 16S analysis. The Granulicatella adiacens had to be identified by 16S because the GAMBIT database did not contain any representatives from the genus Granulicatella.

The 82 species predictions made by GAMBIT described above spanned 51 different species over 29 unique genera. This included correct distinctions between 5 different species of *Staphylococcus* (*S. aureus*, *S. epidermidis*, *S. hominis*, *S. ludgunesis* and *S. saprophyticus*) and 8 different species of *Streptococcus* (*S. agalactiae*, *S. anginosus*, *S. constellatus*, *S. dysgalactiae*, *S. mutans*, *S. pneumoniae*, *S. pyogenes*, *S. salivarius*). Additionally, GAMBIT correctly differentiated one sample of *Shigella sonnei* from six other samples of *Escherichia coli*. The breadth of genera and species correctly predicted by GAMBIT combined with the discriminatory power within genera such as *Staphylococcus* and *Streptococcus* demonstrates the utility of GAMBIT in clinical diagnostics.

## Validation of GAMBIT by the Nevada State Public Health Laboratory

From 2021 to 2022 the Nevada State Public Health Laboratory (NSPHL) performed identifications on 604 isolates. In addition to performing bacterial identification with the normal

**Table 3. Summary of GAMBIT Validations from Alameda County Public Health Laboratories and Nevada State Public Health Laboratories.**

| Validation Set | Isolates | Total Species | Total Genera | False Positives | Species-level calls | Genus-level Calls | No identification |
|---|---|---|---|---|---|---|---|
| **Alameda County Public Health** | 88 | 51 | 29 | 0 | 82 | 4 | 2 |
| **Nevada State Public Health** | 604 | 25 | 15 | 0 | 580 | 27 | 0 |

procedures, all isolates were sequenced on an Illumina sequencing platform using paired end sequencing and identifications generated with GAMBIT. For this validation we compare GAMBIT identifications with the primary identification generated by the NSPHL. NSPHL is a reference laboratory and utilizes multiple methods of microbial identification, including biochemical methods, MALDI-TOF and genomic means (sequencing or probe-based methods). These additional methods were used to determine the identification of the isolate when the initial result was undetermined or called at the genus level only. Thus, NSPHL provides a "gold standard" identification with which to compare GAMBIT identifications.

For all 604 isolates, GAMBIT did not provide any incorrect identifications. This is by design, for the thresholds used by GAMBIT to be conservative and err on the side of genus only or no call rather than produce a false positive identification. Results of this validation are summarized in Table 3. For 518 isolates, GAMBIT and the initial NSPHL identification provided the same identification at the species level. These results spanned 25 species and 15 genera.

For 62 isolates, GAMBIT provided a more specific identification than the initial NSPHL identification (see S8 Table). In two instances, GAMBIT returned the correct genus where the initial NSPHL returned no ID. In another instance, GAMBIT returned the species where the initial NSPHL returned only the genus. There were 49 instances where GAMBIT correctly returned IDs for *Shigella flexneri* and *Shigella sonnei* where the initial NSPHL ID was *Escherichia coli*. The ability of GAMBIT to distinguish *Shigella* from *E. coli* with confidence is a compelling feature for public health applications. Lastly, there were nine instances where GAMBIT called a more specific species variant while the initial NSPHL ID was a more generic call, an example of which is GAMBIT calling *E. hormaechei* when the initial NSPHL ID was *E. cloacae* complex.

For 25 isolates GAMBIT provided only the correct genus where the initial NSPHL analysis identified at the species level (S8 Table). These instances fell broadly into two classes. The first class is where GAMBIT routinely identifies the species well, but a particular isolate or subset of isolates were called at the genus level because of the conservative thresholds. For example, out of 209 *Klebsiella pneumoniae* in this validation set, GAMBIT called 208 at the species level and 1 at the genus level [30]. 8 isolates fall into this category (*Klebsiella*, *Pseudomonas* and *Salmonella* genus-only calls listed in S8 Table). The other class covering 17 isolates from the validation set represent species which are not well-represented in the GAMBIT database and thus are often called at the genus level. For example, of 13 *Enterobacter cloacae* only 3 were called at the species level, the other 10 were called at the genus level. These include *Enterobacter cloacae*, *Mycobacterium chimaera* and *Camplylobacter lari* from this validation set. Future updates to the GAMBIT database could improve performance for these organisms.

Lastly, the Alameda Public Health Laboratory used single-end sequencing in their validation and the Nevada Public Health Laboratory used paired-end sequencing in their validation. Neither method was demonstrably better in performance than the other in our validation sets.

## Tree-building tool based on GAMBIT distances

A benefit of our k-mer based method is the rapid generation of relative relatedness trees utilizing the GAMBIT distance. A user can enter a number of genome assemblies from which they want to generate a relative relatedness tree. GAMBIT's tree-building utility generates all pairwise comparisons amongst those genomes and stores them in a matrix. Hierarchical clustering is then performed on this matrix and a relative relatedness tree is quickly generated. The distance values generated by this process are highly correlated to sequence identity. Because only

a subset of the genome is utilized to calculate the GAMBIT score, this relatedness method is inferior to methods that provide comparisons based on comparing entire genomes (such as Mauve alignments). However, the benefit of this method is the ability to rapidly generate a relatedness tree with dozens of input genomes—something impractical to do if using alignment-based comparison methods.

To validate the usefulness of this tool, we generated a tree from 29 *E. coli* isolates that had been characterized by the Stephens Lab [27] (set 5 in Table 1). All 29 isolates had been assigned to one of five phylogroups based on molecular testing (either group A, B1, B2, D or F). Additionally, MLST subtyping was done on each isolate, which is a more granular division than phylogroups and results in a strain number such as 1193. The results of the relatedness tree are shown in Fig 6. Also shown is a heatmap of pairwise GAMBIT distances for all isolates. Phylogroups are designated by different colored labels and the letter associated with the phylogroup. In all cases members of the same phylogroup cluster together. Additionally, in all cases where isolates share the same MLST, they form tight clusters together. There were four instances of pairs of isolates with the same MLST and three instances of a trio of isolates that have the same MLST (Fig 6). This extension of the GAMBIT scoring system provides a useful tool for researchers and diagnosticians interested in the relatedness of their bacterial isolates.

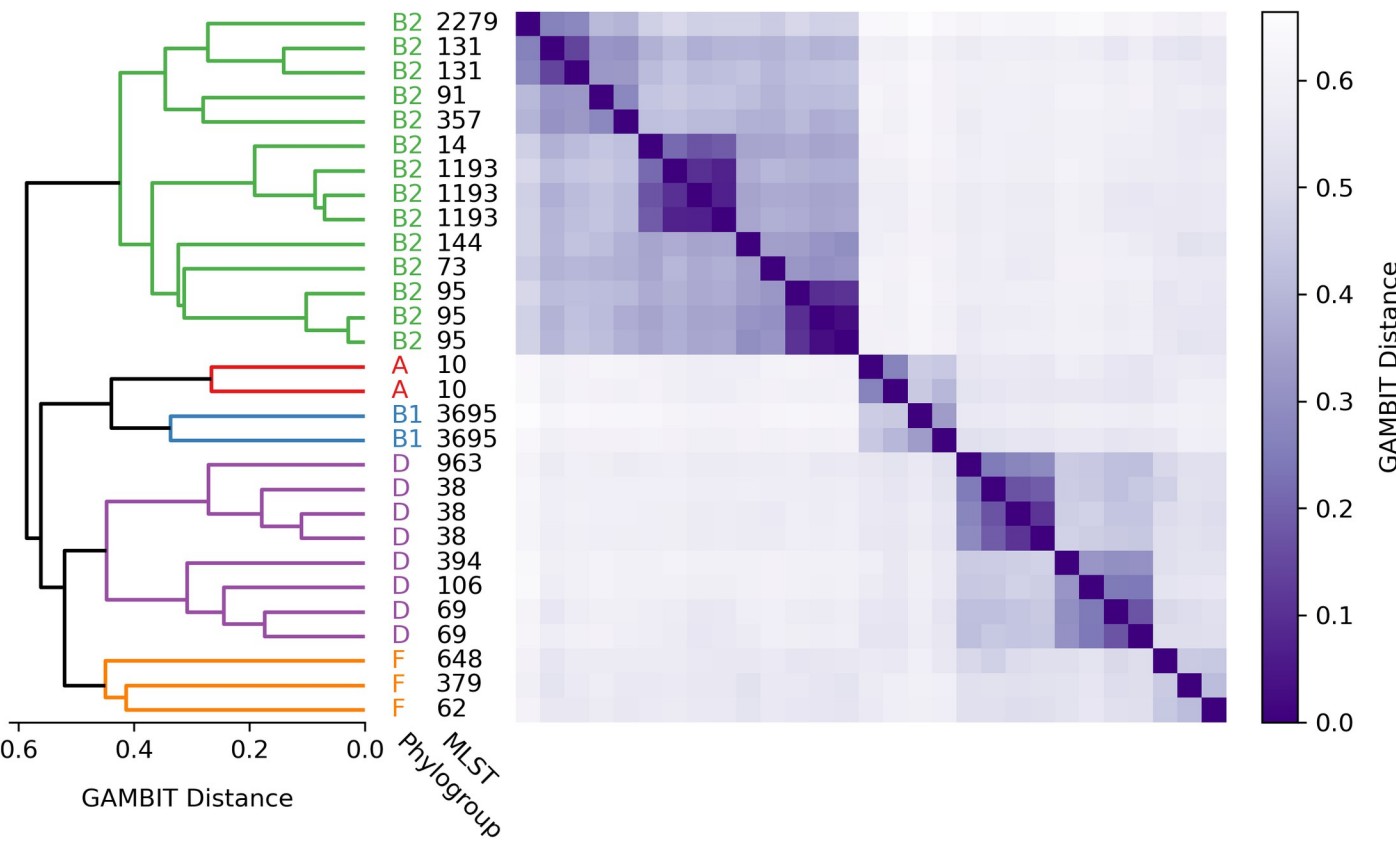

**Fig 6. Phylogenetic tree estimated using GAMBIT distances.** PCR and sequencing were used to perform multilocus sequence typing and determine the phylogroups of 29 *E. coli* genomes provided by the Stephens lab [27] (data set 5). We then calculated all pairwise GAMBIT distances (displayed as a heatmap on the right of the figure) for these genomes using standard parameter values, and used the resulting distance matrix to perform hierarchical clustering using the UPGMA method. The resulting tree groups both phylogroups (colored) and MLST types into clades.

## Discussion

### Utilization of GAMBIT (and whole genome sequencing) as a lab developed test

Since its initial version of development, GAMBIT has been in use at the Alameda County Public Health Laboratory as a laboratory developed test (LDT). After extensive initial validation, the use of whole genome sequencing combined with GAMBIT analysis has been used in over 1500 instances of bacterial identification in a diagnostic setting. The vast majority of instances have resulted in outright identification while occasional instances required use of 16s rRNA or other sequence-based analyses to accomplish complete identification. GAMBIT possesses certain features that make it functional in the diagnostic / CMS (Center for Medicaid/Medicare Services) / CLIA (Clinical Laboratory Improvement Amendments)-certified setting. Firstly, the database that is utilized for identification is highly curated. The curation process identified and removed several genomes in RefSeq that were either misidentified or mislabeled. Such genomes, had they remained in the database, might contribute to a lack of specificity for genome based LDT. Secondarily, the database is locally hosted, in the laboratory. This means that it meets the requirement of existing in a CLIA-certified environment where it can be part of an ongoing quality assurance program. Had the database existed off-site it would have to exist in a CLIA-certified environment. This is because as the specificity component of the test, it plays a crucial role in execution of the test. Thirdly, GAMBIT provides actual and definitive "calls" of identification. The use of BLAST and other genomic similarity tools with online databases typically results in a line listing of similar genomes but without the information necessary to evaluate the significance of these matches with regard to taxonomic identification. GAMBIT instead makes explicit calls based on carefully curated per-taxon classification thresholds. This allows it to function with defined signal to noise parameters that lend extreme confidence to an identity call. Fourthly, GAMBIT provides an output which describes detailed final results and the quantitative aspects of those results that led to the identity call. This means that raw information regarding test performance is available for documentation, which lends itself very well to quality control and quality assurance programs that are required of CLIA certification or other similar accreditation bodies.

The k-mer based algorithm that GAMBIT utilizes takes inspiration from the Aarestrup group [24] and bears similarity to methods from the Phillippy group [25]. The focus for our group was to create a k-mer based tool that could be used diagnostically. To accomplish this, we needed to solve two problems beyond the use of a k-mer based comparison to determine genome similarity. First, we needed a gold-standard reference database that was static so that the result of a query was not dependent on the point in time when the query was run. Public databases, such as Genbank, are constantly updated. Additionally, there is little to no curation of genomes and their species assignment in public databases. As such, they are not as well suited for use in a diagnostic process. Additionally, k-mer comparisons rely on a significant amount of pre-processing of any reference library to achieve the performance gains in terms of speed. We accomplished this by curating and culling all bacterial genomes in RefSeq from date July 1st, 2016 resulting in the necessary static pre-processed database that represented 48,224 genomes from 1414 species. We believe this highly curated list of bacterial genomes with genus and species information will be of general interest to the scientific community. This database will be updated periodically to add new species and additional genomes to existing species.

The second challenge to diagnostic implementation of GAMBIT was developing a set of thresholds that allowed a confident genus and/or species prediction. This is in contrast to existing systems that rely on percent match data and ranking where no particular "call" is made. We leveraged our curated database to perform all pairwise comparisons and used these data to develop thresholds based on both distances within a given taxon and distances to its

nearest sister taxa. We ensured that our method would not call a species or genus if the distance of the inspected genome to the top hit in our database was not within that threshold distance. From Jan 2017 to Sep 2022, Alameda County Public Health Laboratories has used GAMBIT for identification in 28 proficiency testing events covering 118 specimens. During this window, the algorithm has not resulted in a misidentification. We believe that this framework for determining thresholds will lend itself to future improvements by utilizing machine learning approaches to identify k-mers that are most associated with certain genus or species and giving more weight to those k-mers when calling a specific species.

## Benefits of whole genome sequencing for organismal description and molecular epidemiology

While the use of whole genome sequencing for diagnostic microbiology may seem unnecessary, its routine use for this purpose is potent in many respects. Traditional diagnostic bacteriology is a profession that can in many cases require enormous skill and experience. While WGS cannot replace this, it can provide a comprehensive means of identification that can be performed systematically. Additionally, when genomic sequences are generated in the course of identification, the potential to derive medical and public health benefit from the information within identified genomes becomes can be realized. This includes the ability to more rapidly assess cases phylogenetically in furtherance of epidemiology and disease control. It is possible that in the near future, both drug resistance and virulence factors could also be discerned confidently from genomic data, which would potentially eliminate the need for additional steps in the diagnostic workflow.

## Whole genome sequencing in clinical microbiology

Clinical microbiology in public health labs typically falls into three categories: (1) rapid turnaround of identities for unknown infections, (2) reference lab functions for unknown specimens, and (3) surveillance activities around notifiable infections. While WGS is not yet the preferred method for (1), it does provide timely information for functions (2) and (3). In addition, the whole-genome information beyond just the ID provided by methods such as MADLI-TOF provides granularity that is additive to reference and surveillance functions. In addition, having these workflows in place will prepare Public Health Labs for the coming advances in diagnostic WGS. To this point, a letter to the editor of the New England Journal of Medicine [31] demonstrated a diagnosis time from blood to human WGS of 7 hours and 18 minutes. As these technologies continue to improve microbial WGS is likely to provide rapid identifications not just for reference and surveillance activities but also rapid turnaround identifications. Having infrastructure and training in place for WGS will prepare diagnostic labs for the coming technological advances.

## GAMBIT distance compared to average nucleotide identity

Average Nucleotide Identity (ANI) has been the benchmark for nucleic acid comparisons. Our comparisons in this paper reveal a nearly monotonic relationship between GAMBIT distance and ANI. Because our taxonomic classification process only uses this value for order-based comparisons (selecting the closest reference genome to a query and comparing this distance to a set threshold), classification results using our system should be nearly equivalent to those that would result from using the same method with the ANI values themselves. As shown in S2 Fig, GAMBIT distance extends more broadly in taxonomic structure and has peformance advantages over ANI in implementation. These factors make GAMBIT a significant improvement over ANI for microbial classification.

## Materials and methods

### Obtaining clinical samples from the alameda county public health laboratory

Bacterial organisms were obtained and selected in either of two ways: a) isolates generated or received at the Alameda County Public Health Laboratory through normal diagnostic requests from jurisdictional hospitals were identified through validated and FDA cleared methods. The primary method was API Strip (Biomerieux, Durham, NC). No instances were detected in the course of the study that demonstrated improper identity by way of API strip testing. Isolates so identified were confirmed for genus and species identity by reference laboratory (California Department of Public Health) using the same or alternative methods; another means, b) included the use of bacterial isolates provided by proficiency testing (College of American Pathologists). Isolates received for quality assessment were saved by freezing the acquired isolates and subsequently analyzed through whole genome sequencing and then by GAMBIT. Consensus identification was verified by report from the College of American Pathologists and compared to GAMBIT results.

### Obtaining clinical samples from the Nevada State Public Health Laboratory

Bacterial organisms were obtained from routine specimen submissions at the Nevada State Public Health Laboratory. The organisms were identified through validated and FDA cleared methods. Most bacterial identification was performed using BD Phoenix M50 and MALDI--TOF (Siruis, Bruker). Mycobacterium were identified using AccuProbe (Hologic). Additional species identification was performed on enteric bacteria using ANI (Average Nucleotide Identity) in BioNumerics 7.6 (Applied Maths).

### Whole genome sequencing from the Alameda County Public Health Laboratory

Isolates were collected from culture plates and were subject to genomic DNA extraction using the MagNAPure Compact (Roche, Switzerland). The Bacterial lysis procedure was used, according to the package insert. Extracted genomes were assessed for DNA concentration by Qbit fluorescent analyzer. Library preparation was performed by using the Nextera XT (Illumina, La Jolla, CA) with the following alteration to the package insert protocol: final library quantitative normalization was not performed by using the bead-based method in the procedure. Instead, final library concentrations per specimen were determined by fluorimetry and were equalized by dilution in water. Sequencing was performed on the MiSeq instrument and included single-end reads, using 2X150 sequencing kits. Sequencing data was processed with Theiagen's bacterial genomics workflow, v 0.4.0, (see section on software availability), using SPADES (v3.14.1) [32] for de-novo assembly through the Shovill pipeline (v 1.1.0) [33]. A single sample did not produce an acceptable assembly with this method and was instead assembled using Tadpole (v36.38) [34] (see S3 Table).

### Whole genome sequencing from the Nevada State Public Health Laboratory

Isolates were collected from culture plates and were subject to genomic DNA extraction using DNeasy Blood and Tissue kit on a QiaCube instrument (Qiagen). Extracted genomes were assessed for DNA concentration by Qubit 3 Fluorometer (Invitrogen). Library preparation was performed by using the Illumina DNA Prep kit (Illumina). Paired-end sequencing was

performed, using 2X150 MiniSeq Mid-Output sequencing kits (Illumina). Data processing and de-novo assembly were performed as in the previous section.

## GAMBIT signature generation and genomic distance metric

A GAMBIT signature is a compressed representation of a genome sequence which supports efficient calculation of the GAMBIT genomic distance metric. It is defined as the set of k-mers present in the genome which occur immediately following a fixed prefix sequence. The signature depends on the value of two parameters—k (a positive integer) and the prefix. The genome database GAMBIT uses for taxonomic classification uses the values $k = 11$ and prefix = ATGAC. The software supports calculation of signatures from genome assembly files in FASTA format.

Internally, the GAMBIT software library represents k-mers using integer indices in the range $[0 .. 4^k - 1]$. Indices are derived using the alphabetical ordering of all $4^k$ k-mers for the given value of $k$. Thus the 5-kmer AAAAA has index 0 and TTTTT has index $4^5 - 1 = 1023$. Full genome signatures are stored as arrays of k-mer indices in sorted order using the smallest possible unsigned integer data type. Reference genome signatures in the GAMBIT database consist of 7,000 32-bit integers on average, requiring only 28 kilobytes of storage. The software defines a binary file format which is capable of storing any number of genome signatures alongside arbitrary metadata. The GAMBIT distance between two genomes is calculated as the Jaccard distance between the k-mer sets that comprise their signatures. Naturally, the two signatures must have been calculated using identical parameter values (k and prefix sequence).

The GAMBIT signature is analogous to the concept of a "sketch" in the Mash tool [25] in that it is a representation of the k-mer content of a genome which supports calculation of a genomic distance metric based on the Jaccard distance. The key difference is in how the two tools accomplish this task with a level of computational efficiency that supports running tens or hundreds of thousands of such comparisons in a short time on modest hardware. Mash uses the MinHash [35] technique to create a highly compressed "sketch" of the full k-mer content of a genome, which can be used to approximate the true Jaccard distance between the sets represented by two such sketches. By contrast GAMBIT performs a subsampling of the genome's full k-mer set and calculates the Jaccard distance between two of these subsets directly. Despite its relative simplicity, we show genomic distances calculated using our method still correlate very highly to ANI.

## GAMBIT taxonomic classification

The GAMBIT database used for classification consists of precalculated signatures for 48,224 reference genomes along with additional genome metadata and a taxonomy tree. Taxonomy information is derived from the NCBI taxonomy database but restricted to the genus and species ranks and subject to additional manual curation. Each reference genome is assigned to a taxonomy node, typically of species rank. In a small number of cases where manual editing of the taxonomy structure was required, genomes were assigned to artificial nodes below the species level. Genus and species nodes are assigned a threshold distance.

The classification process takes as its input an assembled query genome in FASTA file format. GAMBIT first calculates the signature of the query sequence and then the distance from the query to each reference genome signature. From this it selects the reference genome with the minimum distance to the query, and compares this distance to the threshold of the reference genome's assigned species. If the distance is less than this threshold then GAMBIT will classify the query as the given species. Otherwise GAMBIT will ascend the taxonomy tree from the species node to its parent genus node and repeat the same process, possibly returning a

genus-level classification. If the query does not fall within the genus' threshold distance GAMBIT will be conservative and report the query as "unknown." Regardless of the level of classification made, GAMBIT will always report the closest reference genome along with its taxonomic assignment and distance to the query.

## GAMBIT tree generation

GAMBIT includes a function to estimate a relatedness tree for a given set of genome assemblies. It calculates pairwise distances between all input genomes and then performs hierarchical clustering using the UPGMA method. The resulting tree can be output in Newick format or visualized as a dendrogram.

## ANI score generation

ANI scores in this manuscript were generated using the FastANI tool [21] (version 1.33) with default parameter values (k-mer size 16 and fragment length 3000).

## Performance benchmarks

Performance of the full GAMBIT taxonomic classification process was measured on a Dell XPS laptop with a six-core Intel i7-9750H processor and 16 gigabytes of memory, with three replicates run for each genome set 1–4.

## Code availability

GAMBIT source code has been made publicly available under the GNU AGPL 3.0 license and is hosted as a Git repository: https://github.com/jlumpe/gambit/. It is implemented in Python 3 and makes significant use of the BioPython [36], Numpy [37], and Pandas [38] libraries. It may be used either as a command line tool or as an importable Python library to access its low-level functionality in third-party scripts or software. Documentation for both use cases is hosted at: https://gambit-genomics.readthedocs.io/.

GAMBIT is available to install through the Bioconda [39] channel of the popular Conda package manager: https://bioconda.github.io/recipes/gambit/README.html. The State Public Health Bioinformatics (StaPH-B) consortium also maintains Docker container images of GAMBIT releases accessible through the StaPH-B DockerHub (https://hub.docker.com/r/staphb/gambit/) and Quay (https://quay.io/repository/staphb/gambit) repositories. Dockerfiles for these images have been made available through the StaPH-B docker-builds GitHub repository: https://github.com/StaPH-B/docker-builds.

Additionally, Theiagen Genomics have created end-to-end workflows for de-novo assembly, quality control, and taxonomic identification of bacterial NGS data which incorporate GAMBIT functionality. They are implemented in WDL (workflow description language) and have been made available through the Theiagen Genomics Public Health Bacterial Genomics (PHBG) Dockstore Collection: https://dockstore.org/organizations/Theiagen/collections/PublicHealthBacterialGenomics/. While these workflows can be run locally or on an HPC system at the command-line, utility was optimized for public health end-users with limited programming or bioinformatics experience through Terra, a bioinformatics web application developed by the Broad Institute of MIT and Harvard in collaboration with Microsoft and Verily Life Sciences: https://app.terra.bio/.

Reproducible code to generate all figures and major analyses in this manuscript has been made available as an automated workflow using the Snakemake [40] workflow management system: https://github.com/jlumpe/gambit-publication. In addition to the libraries mentioned

above, the workflow uses Matplotlib [41] and Seaborn [42] to generate figures. The workflow uses release 1.0.0 of the GAMBIT software.

## Supporting information

**S1 Fig. Spearman correlation between GAMBIT distance and ANI for a larger ensemble of parameter values.** Subplots represent different choices of prefix sequence for the GAMBIT distance metric. Columns correspond to eight 7-mer base sequences (our default ATGAC plus two random nucleotides and seven fully random sequences). Rows correspond to truncations of these base sequences to 4, 5, 6, and 7 nucleotides. As in Fig 2 each subplot shows the absolute value of the Spearman correlation between GAMBIT distance and ANI vs value of the k parameter for all genome pairs in data sets 1–4.
(PNG)

**S2 Fig. GAMBIT similarities of genome pairs split based on whether ANI values could be estimated.** Shown are distributions of GAMBIT similarity values (one minus the GAMBIT distance) for all pairs of genomes in sets 1–4, split based on whether the FastANI tool did (blue) or did not (green) report an ANI value for the pair. FastANI's cutoff of approximately 80% corresponds to a GAMBIT distance of approximately 0.99 (similarity 0.01), with very little overlap between the two groups. Note that FastANI reported a value for all pairs in set 1.
(PNG)

**S1 Table. Genome assemblies in data set 1.**
(XLSX)

**S2 Table. Genome assemblies in data set 2.**
(XLSX)

**S3 Table. Genome assemblies in data set 3.**
(XLSX)

**S4 Table. Genome assemblies in data set 4.**
(XLSX)

**S5 Table. Reference genomes in the GAMBIT database.**
(XLSX)

**S6 Table. Taxa in the GAMBIT database.** "Threshold" is the distance threshold used for classification. In all cases the threshold is less than "min inter", the "overlap" column indicates cases in which this value was less than or equal to "max intra" (these terms are described in the section titled "Establishing confidence thresholds for classification"). Taxa with a blank NCBI ID and/or a rank of "none" were manually added or restructured as part of the curation process. Taxa with a threshold of zero are used for reporting results only, classification is performed using their children.
(XLSX)

**S7 Table. GAMBIT prediction results for proficiency test isolates from 2016–2020 generated at the Alameda County Public Health Laboratories which are the data in set 3.**
(XLSX)

**S8 Table. GAMBIT predictions compared to initial characterization by Nevada State Public lic Health Laboratories.**
(XLSX)

## Acknowledgments

We thank all employees of both the Alameda County Public Health Laboratory and Nevada State Public Health Laboratory involved with the intake and processing of pathogenic microbial isolates for identification and characterization. Without their efforts this study would not be possible.

## Author Contributions

**Conceptualization:** Jared Lumpe, David Hess, Mark Pandori.

**Data curation:** Jared Lumpe, Kevin Libuit, David Hess.

**Formal analysis:** Jared Lumpe, Andrew Gorzalski, Vici Varghese, David Hess.

**Funding acquisition:** Craig Stephens, Joel Sevinsky, David Hess, Mark Pandori.

**Investigation:** Jared Lumpe, Vici Varghese, Tyler Lloyd, David Hess, Mark Pandori.

**Methodology:** Jared Lumpe, Andrew Gorzalski, Kevin Libuit, Eileen Wagner, Joel Sevinsky, David Hess, Mark Pandori.

**Project administration:** David Hess, Mark Pandori.

**Resources:** Craig Stephens, David Hess, Mark Pandori.

**Software:** Jared Lumpe, Lynette Gumbleton, Kevin Libuit, Farid Tadros, Joel Sevinsky, David Hess.

**Supervision:** David Hess.

**Validation:** Jared Lumpe, Lynette Gumbleton, Andrew Gorzalski, Vici Varghese, Tyler Lloyd, Tyler Arsimendi, Eileen Wagner, Craig Stephens, David Hess, Mark Pandori.

**Writing – original draft:** Jared Lumpe, Kevin Libuit, Vici Varghese, David Hess, Mark Pandori.

**Writing – review & editing:** Andrew Gorzalski, Kevin Libuit, Joel Sevinsky, David Hess, Mark Pandori.

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
