## [Decision Letter · Decision Letter 0]

7 Sep 2022

PONE-D-22-20500GAMBIT (Genomic Approximation Method for Bacterial Identification and Tracking): A methodology to rapidly leverage whole genome sequencing of bacterial isolates for clinical identificationPLOS ONE

Dear Dr. Hess,

Thank you for submitting your manuscript to PLOS ONE. After careful consideration, we feel that it has merit but does not fully meet PLOS ONE’s publication criteria as it currently stands. Therefore, we invite you to submit a revised version of the manuscript that addresses the points raised during the review process.

Your manuscript has been reviewed and a minor revision is suggested. Please follow the reviewer comments and make all necessary revision.

We look forward to receiving your revised manuscript.

Kind regards,

Chih-Chieh Chen, Ph.D.

Academic Editor

PLOS ONE

Journal Requirements:

"This work was supported in part by the National Institutes of Health, grant R15AI130816-01A1. We thank the College of Arts and Sciences at Santa Clara University for supplemental funding."

"No. The funders had no role in study design, data collection and analysis, decision to publish, or preparation of the manuscript."

Additional Editor Comments:

Your manuscript has been reviewed and a minor revision is suggested. Please follow the reviewer comments and make all necessary revision.

Reviewers' comments:

Reviewer's Responses to Questions

**Comments to the Author**

1. Is the manuscript technically sound, and do the data support the conclusions?

Reviewer #1: Yes

Reviewer #2: Yes

2. Has the statistical analysis been performed appropriately and rigorously? 

Reviewer #1: I Don't Know

Reviewer #2: I Don't Know

3. Have the authors made all data underlying the findings in their manuscript fully available?

Reviewer #1: Yes

Reviewer #2: Yes

4. Is the manuscript presented in an intelligible fashion and written in standard English?

Reviewer #1: Yes

Reviewer #2: Yes

5. Review Comments to the Author

Reviewer #1: This is a well written manuscript that has described a method to identify bacterial species using whole genome sequencing data and utilizes a kmer approach that the authors show to be rapid and reliable. The authors also rightly identified problems in current databases and their curation would be much appreciated. This manuscript will be of interest to the sequencing community, but I wonder if it would make much of an impact in clinical microbiology.

Current methods used to identify bacterial species in the clinical micro lab include biochemical tests which are notoriously unreliable, which is why this has been replaced by MALDI-TOF. The latter method is fairly reliable although there are many genera which it has problems with, and it is a pity that the authors of this paper did not address more of these genera (Acinetobacter baumannii complex, Enterobacter cloacae complex, Klebsiella pneumonia complex, Streptococcus spp.) and instead looked at easier to identify bacteria like E. coli. I was also surprised that API strips were reportedly used for species identification from one testing laboratory. How did they perform?

Furthermore, sequencing simply takes too much time to be used as a routine method for species identification. Certainly in my experience, whole genome sequencing is only used in rare cases where a definitive method for identification is requested, although as stated above, given the time it takes to get a result, the impact of the identification on patient treatment is rather limited. The time to result should be discussed in the manuscript.

Was there a difference in results between the single- and paired-end reads?

How did GAMBIT compare to other databases such as JSpecies?

In Table 2, you state for example Enterobacter hormaechei. Can you identify to the subspecies? This is important information as in our institute we find their various different subspecies. Similarly, can you delineate the Klebsiella subspecies? Acinetobacter is another notoriously difficult genus to identify some of the species. The data presented in this manuscript would be enriched if you could add some more of these different sequences to your database and test them.

Reviewer #2: I appreciate the chance to review this manuscript which describes a tool for k-mer based bacterial identification from WGS data. The manuscript uses an approach of specifying a k-mer prefix of ATGAC tagged on to 11-mers to reduce the overall search space and database complexity to speed processing time, as well as a curated database to help insure accuracy of results.

I have a few questions and comments which i would like to see addressed.

1) lines 108-116 - this seems to be a mix of methods and discussion under the heading of results. It is a bit of a gray area since the manuscript is describing software but still, I think an attempt should be made to keep the information in the relevant sections and subheadings.

2) line 154 - 157 if this represents a belief then it seems better characterized as discussion rather than a result.

3) line 399 - in noting difficulty with Klesbiella, there are several publications which detail Klebsiella which are misidentified in Genbank as well as difficulty in the classification of Klebsiella by MALDI and other typing methods, none of which are cited here but which might be of interest to the authors.

4) line 491 - The statement that starts "to date, in routine... has not resulted in a misidentification." this would be far more useful and powerful if it were quantitatively stated, ie. "over X period of time, and Y number of samples tested, there has not been a misidentification ..."

5) Lines 499 - 519 are perhaps the most controversial and problematic of the manuscript for several reasons which I will detail below. While I am a champion for the use and utility of WGS in clinical diagnostic microbiology as well as public health, the authors do themselves a disservice by the strident approach taken in these points. In particular, in point 1, I fail to see how this "eliminates the need" for bacteriologists to perform identifications when at several points in the results the authors point out the failures and shortcomings of their method (lin 311-313, line 316, line 294, line 399, and lines 350-352). Indeed, how will we identify bacteria when the curated reference database fails and who will work to identify what needs to be added to the database in the future? Given the critical shortage of medical technologists and their utility at multiple steps in the process, the suggestion that this algorithm + WGS eliminates the need for them is offensive to this reader. One might argue that their numbers are decreasing in part because they are not seen as valued members of the clinical team and due to predictions of their replacement by automation which have proven premature. In the second point, line 507, virulence and drug resistance can not necessarily be "ascertained immediately" - especially for some antimicrobial resistance phenotypes which derive from multiple de novo SNVs or other changes, not just resistance gene presence / absence. Also, clinicians have difficultly interpreting resistance gene data and how is this addressed? There are publications which cover these topics none of which are cited. In line 511, relatedness can be deceiving in clinical environments. For instance there can be patients admitted around the same time with similarly related bacterial infections acquired elsewhere, and bacteria vary at differing rates depending on species, complicating analysis. These topics should be addressed. In point 4, the statement "Normal ... diagnostics provide no insight into how organisms change over time" is simply not true. Antimicrobial resistance data is followed by microbiologists and infection control, and yearly antibiograms are produced, to track and follow changes in organisms over time and in fact can be tracked in a single patient to see how resistance is acquired over time which is relevant to clinical care. Line 517 would reconsider the use of the word "mating" also there are numerous mentions in the last section of the relevance to public health in a section devoted to the utility of sequencing to diagnostics, in particular.

In sum total for this section (499-519), I would urge the authors to reconsider their statements and their impact on readers and fellow healthcare professionals, and temper their enthusiasm for the technique with a dash of their previously acknowledged limitations many of which have been previously described in the literature but are not cited or mentioned in the manuscript.

6) Methods - software needs versions mentioned throughout. I appreciate the thorough documentation and provision of the code online.

Figures - Legends are well detailed, the figures in the supplied PDF appear grainy / aliased / pixelated. Otherwise fine.

One final comment, I feel like this is a topic which has been published on multiple times in the past decade and that a variety of literature exists to support the authors discussions but which is not cited in the manuscript. I would encourage the authors to make sure that all claims are either supported by the data presented or the relevant literature is cited.

6. PLOS authors have the option to publish the peer review history of their article (what does this mean?). If published, this will include your full peer review and any attached files.

Reviewer #1: No

Reviewer #2: No

---

## [Author Response · Author response to Decision Letter 0]

10 Oct 2022

Response to editor and reviewer comments for “GAMBIT (Genomic Approximation Method for Bacterial Identification and Tracking): A methodology to rapidly leverage whole genome sequencing of bacterial isolates for clinical identification” (submission ID number: PONE-D-22-20500)

Thank you for the reviews of “GAMBIT (Genomic Approximation Method for Bacterial Identification and Tracking): A methodology to rapidly leverage whole genome sequencing of bacterial isolates for clinical identification” (submission ID number: PONE-D-22-20500). These reviews helped us strengthen the manuscript and we are pleased to submit this revised version. Below we detail the changes that were made to the manuscript in response to specific points made by the reviewers.

Editor comments

Editor comment:

1) Please ensure that your manuscript meets PLOS ONE's style requirements, including those for file naming. The PLOS ONE style templates can be found at https://journals.plos.org/plosone/s/file?id=wjVg/PLOSOne_formatting_sample_main_body.pdf and https://journals.plos.org/plosone/s/file?id=ba62/PLOSOne_formatting_sample_title_authors_affiliations.pdf

Author response:

We have updated the style of the document to match PLoS guidelines, particularly with respect to the author and affiliations list and figure and table references. Figure captions and tables have been moved to/inserted at the correct locations.

Editor comment:

2) We note that the grant information you provided in the ‘Funding Information’ and ‘Financial Disclosure’ sections do not match. When you resubmit, please ensure that you provide the correct grant numbers for the awards you received for your study in the ‘Funding Information’ section.

Author response:

Responses have been added to the new cover letter.

Editor comment:

3) Thank you for stating the following in the Acknowledgments Section of your manuscript: "This work was supported in part by the National Institutes of Health, grant R15AI130816-01A1. We thank the College of Arts and Sciences at Santa Clara University for supplemental funding."We note that you have provided funding information that is not currently declared in your Funding Statement. However, funding information should not appear in the Acknowledgments section or other areas of your manuscript. We will only publish funding information present in the Funding Statement section of the online submission form. 

Please remove any funding-related text from the manuscript and let us know how you would like to update your Funding Statement. Currently, your Funding Statement reads as follows: "No. The funders had no role in study design, data collection and analysis, decision to publish, or preparation of the manuscript." Please include your amended statements within your cover letter; we will change the online submission form on your behalf.

Author response:

Responses have been added to the new cover letter.

Editor comments:

4) We note that you have indicated that data from this study are available upon request. PLOS only allows data to be available upon request if there are legal or ethical restrictions on sharing data publicly. For more information on unacceptable data access restrictions, please see http://journals.plos.org/plosone/s/data-availability#loc-unacceptable-data-access-restrictions.

Author response:

Responses have been added to the new cover letter.

Editor comments:

5) PLOS requires an ORCID iD for the corresponding author in Editorial Manager on papers submitted after December 6th, 2016. Please ensure that you have an ORCID iD and that it is validated in Editorial Manager. To do this, go to ‘Update my Information’ (in the upper left-hand corner of the main menu), and click on the Fetch/Validate link next to the ORCID field. This will take you to the ORCID site and allow you to create a new iD or authenticate a pre-existing iD in Editorial Manager. Please see the following video for instructions on linking an ORCID iD to your Editorial Manager account: https://www.youtube.com/watch?v=_xcclfuvtxQ

Author response:

Dr. David Hess has registered under ORCID ID 0000-0003-2821-1021, which will be updated upon resubmission.

Editor comments:

Author response:

References have been reviewed and several citations were added in the revised manuscript.

Revisions to submitted manuscript

● Updated style to match PLoS guidelines

○ Reformatted author list and affiliations.

○ Moved table and figure captions according to where they were first mentioned.

○ Formatted figure and table references in text.

● Data changes

○ A single genome was removed from set 4 due to failing QC, bringing the count down from 605 to 604. All figures and data depending on this data were regenerated using the same automated workflow as before. This resulted in some summary data (Spearman correlation and percent reported) for panel 4 of figure 1 changing slightly at the 3rd significant digit, which did not affect interpretation of the results.

● Corrections

○ Some numbers mentioned in the section “GAMBIT genomic distance metric correlates with sequence identity” were slightly off. In the sentence “Spearman correlation was high in all four data sets …”, the numbers for data sets 3 and 4 should have been -.969 and -.978 respectively, not -.978 and -.949. The second figure was then updated to -.979 due to the removed genome mentioned above.

○ In the next sentence, the percentage of scores reported by FastANI were all updated to a consistent 3 significant figures. The removal of a genome from set 4 did technically change the last figure but not enough to change the digits at this precision.

● Updated content:

○ Added section titled “Whole Genome Sequencing in Clinical Microbiology” to discussion to address comment from Reviewer #1.

○ Added a few sentences to the end of the results section “Validation of GAMBIT by the Nevada State Public Health Laboratory” to address comments by Reviewer #1.

○ Replaced most of “Benefits of Whole Genome Sequencing for Organismal Description and Molecular Epidemiology” to address Reviewer #2 comment 5.

○ Added a sentence to “Obtaining clinical samples from the Alameda County Public Health Laboratory” to address question about API strips from Reviewer #1.

○ Added most of paragraph 2 in “Establishing confidence thresholds for classification” to clarify reasoning behind our method of determining confidence thresholds.

○ Added software versions to address Reviewer #1 comment 6.

○ Updated “Data availability”

○ Updated Acknowledgements section.

● Figures and tables

○ Added description for S1 Fig.

○ Rewrote/added to descriptions for S2 Fig and S6 (previously S4) Table.

○ Inserted two new supplemental data tables at numbers 3 and 4 which list genomes in data sets 3 and 4, including biosample/bioproject numbers to comply with data availability requirements. Supplemental tables previously labeled 3-6 have had their labels incremented by two.

● Corrected first author affiliations

● Corrected wording in numerous places to increase clarity without substantially changing content. This is most significant in the following locations:

○ “GAMBIT distances are robust with regard to parameter choice” The first paragraph was completely rewritten but represents the same information.

○ “Establishing confidence thresholds for classification”, except for the small amount of new content mentioned above.

○ Fig 5 caption. Collapsed common elements of subplots A and B into a few sentences at the start.

● Other miscellaneous:

○ Changed title of section ‘Establishing confidence thresholds for generating “resulting identifications” vs. “suggested identifications”’ to “Establishing confidence thresholds for classification”.

○ Fixed/updated URLs under “Code availability”.

○ Minor changes to labels of S1, 2, and 5 (previously 3) Tables.

○ Truncated numbers in Table 2 to four decimal places.

○ Removed sentences erroneously copied from “Whole genome sequencing from the Alameda County Public Health Laboratory” and pasted in “Curated, High-Quality Database construction based on RefSeq”.

● Additional references:

○ 14. Rodriguez-R LM, Gunturu S, Harvey WT, Rosselló-Mora R, Tiedje JM, Cole JR, et al. The Microbial Genomes Atlas (MiGA) webserver: taxonomic and gene diversity analysis of Archaea and Bacteria at the whole genome level. Nucleic Acids Res. 2018;46: W282–W288.

○ 18. Punina NV, Makridakis NM, Remnev MA, Topunov AF. Whole-genome sequencing targets drug-resistant bacterial infections. Hum Genomics. 2015;9: 19.

○ 22. Balloux F, Brønstad Brynildsrud O, van Dorp L, Shaw LP, Chen H, Harris KA, et al. From Theory to Practice: Translating Whole-Genome Sequencing (WGS) into the Clinic. Trends Microbiol. 2018;26: 1035–1048.

○ 30. Wyres KL, Lam MMC, Holt KE. Population genomics of Klebsiella pneumoniae. Nat Rev Microbiol. 2020;18: 344–359.

○ 31. Gorzynski JE, Goenka SD, Shafin K, Jensen TD, Fisk DG, Grove ME, et al. Ultrarapid Nanopore Genome Sequencing in a Critical Care Setting. N Engl J Med. 2022;386: 700–702.

Responses to reviewer 1

Reviewer comments:

(1) This is a well written manuscript that has described a method to identify bacterial species using whole genome sequencing data and utilizes a kmer approach that the authors show to be rapid and reliable. The authors also rightly identified problems in current databases and their curation would be much appreciated. This manuscript will be of interest to the sequencing community, but I wonder if it would make much of an impact in clinical microbiology.

(2) Current methods used to identify bacterial species in the clinical micro lab include biochemical tests which are notoriously unreliable, which is why this has been replaced by MALDI-TOF. The latter method is fairly reliable although there are many genera which it has problems with, and it is a pity that the authors of this paper did not address more of these genera (Acinetobacter baumannii complex, Enterobacter cloacae complex, Klebsiella pneumonia complex, Streptococcus spp.) and instead looked at easier to identify bacteria like E. coli. I was also surprised that API strips were reportedly used for species identification from one testing laboratory. How did they perform?

(3) Furthermore, sequencing simply takes too much time to be used as a routine method for species identification. Certainly in my experience, whole genome sequencing is only used in rare cases where a definitive method for identification is requested, although as stated above, given the time it takes to get a result, the impact of the identification on patient treatment is rather limited. The time to result should be discussed in the manuscript.

Author response to reviewer points 1-3:

Thank you for your concerns about the impact on clinical microbiology and the turnaround time. We have added a section in the discussion labeled “Whole Genome Sequencing in Clinical Microbiology” to address these issues. In brief, public health clinical microbiology labs serve three major functions, (1) rapid turnaround of identities for unknown infections, (2) reference lab functions for unknown specimens, and (3) surveillance activities around notifiable infections. While WGS is not yet the preferred method for (1), it does provide timely information for functions (2) and (3). In addition, the whole-genome information beyond just the ID provided by MADLI-TOF, provides granularity that is additive to reference and surveillance functions. We thank the reviewer for allowing us to clarify the role of WGS in public health laboratories.

Lastly, the reviewer comments on the paper not addressing genera such as Acinetobacter, Enterobacter, Klebsiella and Streptococcus. In our two validation studies presented (sections “Validation of GAMBIT by the Alameda County Public Health Laboratory” and “Validation of GAMBIT by the Nevada State Public Health Laboratory” we have the following number if each of those genera in our validation sets: Acinetobacter (40), Enterobacter (21), Klebsiella (226) and Streptococcus (28). We feel these test sets are adequate and represent well the diversity in public health microbiology.

Reviewer comments:

(4) Was there a difference in results between the single- and paired-end reads?

Author response:

Because of the k-mer based method where we subsampled only a few percent of the genome, our method is not subject to the constraints of “finished” genomes. We have added to the results section that the Alameda Validation Set was performed with single-end sequencing and that the Nevada validation set was performed with paired-end sequencing (previously this was only in the methods). In addition we added the following sentence to the manuscript:

“Lastly, the Alameda Public Health Laboratory used single-end sequencing in their validation and the Nevada Public Health Laboratory used paired-end sequencing in their validation. Neither method was demonstrably better in performance than the other in our validation sets.” We thank the reviewer for allowing us to make this comparison more directly in the manuscript.

Reviewer comments:

(5) How did GAMBIT compare to other databases such as JSpecies?

Author response:

To the authors’ reading, JSpecies is a tool to find similar genomes in a curated reference database on the basis of ANI (or another genomic similarity metric, the Tetranucleotide Correlation Index). The authors have presented exhaustive comparisons to ANI in both the main manuscript and in the supplemental materials. As ANI is the benchmark “gold standard” for such comparisons, we feel this should stand as the appropriate comparison. We provide all necessary information for investigators interested in performing comparisons between GAMBIT and any other methods (including JSPECIES) and welcome such investigations. There are numerous methods that have been devised and an exhaustive comparison seems beyond the scope of this publication. However, we have included JSPECIES as an additional reference in the publication and thank the reviewer for bringing this to our attention.

Reviewer comments:

(6) In Table 2, you state for example Enterobacter hormaechei. Can you identify to the subspecies? This is important information as in our institute we find their various different subspecies. Similarly, can you delineate the Klebsiella subspecies? Acinetobacter is another notoriously difficult genus to identify some of the species. The data presented in this manuscript would be enriched if you could add some more of these different sequences to your database and test them.

Author response:

We thank the reviewer for noting subspecies calls for GAMBIT. The initial version of GAMBIT focuses only on genus and species calls. But, the information in Figure 6 demonstrates clearly that GAMBIT can make calls below the species level. In fact, sub-group calls are made for 17 species in the database, though we return only species calls (see section “Establishing confidence thresholds for classification” and Figure 5). We are currently working on updates to GAMBIT that will not only bring in additional genera and species to the database but will introduce sub-species calls. We acknowledge the reviewer for seeing where our scholarship is headed, but respectfully feel this addition is outside the scope of this current manuscript.

Responses to reviewer 2

Reviewer comments:

1) lines 108-116 - this seems to be a mix of methods and discussion under the heading of results. It is a bit of a gray area since the manuscript is describing software but still, I think an attempt should be made to keep the information in the relevant sections and subheadings.

Author response:

We understand and thank the reviewer for the comment lines 108 to 116. There are areas of this manuscript that blur the line between methods, results and discussion and have made edits to this effect in other portions of the manuscript. We feel this context is crucial to understanding the main point of the paper and suggest that these lines remain where they are in the manuscript to center the reader on the focus of the manuscript.

Reviewer comments:

2) line 154 - 157 if this represents a belief then it seems better characterized as discussion rather than a result.

Author response:

As written in the original submission, we agree with the reviewer that the text suggests a place in the discussion. Our word choice was poor and did not properly reflect the logical, definitive flow from these results. We have rewritten this paragraph of the results and feel the statement belongs with the results.

Reviewer comments:

3) line 399 - in noting difficulty with Klesbiella, there are several publications which detail Klebsiella which are misidentified in Genbank as well as difficulty in the classification of Klebsiella by MALDI and other typing methods, none of which are cited here but which might be of interest to the authors.

Author response:

Thank you for the note on Klebsiella. We are adding a comprehensive Nature Review Microbiology on the genomics of Klebsiella for additional context (citation below). As the focus of this paper is whole-genome sequencing for species identification, we don’t feel this section of the manuscript is appropriate to compare other specific techniques such as MALDI for Klebsiella, though these are interesting discussions and would be more appropriate for a manuscript focused on Klebsiella.

Wyres, K.L., Lam, M.M.C. & Holt, K.E. Population genomics of Klebsiella pneumoniae. Nat Rev Microbiol 18, 344–359 (2020). https://doi.org/10.1038/s41579-019-0315-1

Reviewer comments:

4) line 491 - The statement that starts "to date, in routine... has not resulted in a misidentification." this would be far more useful and powerful if it were quantitatively stated, ie. "over X period of time, and Y number of samples tested, there has not been a misidentification ..."

Author response:

We have rewritten the statement as follows per the reviewers suggestion:

From Jan 2017 to Sep 2022, Alameda County Public Health Laboratories has used GAMBIT for identification in 28 proficiency testing events covering 118 specimens. During this window, the algorithm has not resulted in a misidentification.

Reviewer comments:

5) Lines 499 - 519 are perhaps the most controversial and problematic of the manuscript for several reasons which I will detail below. While I am a champion for the use and utility of WGS in clinical diagnostic microbiology as well as public health, the authors do themselves a disservice by the strident approach taken in these points.

a) In particular, in point 1, I fail to see how this "eliminates the need" for bacteriologists to perform identifications when at several points in the results the authors point out the failures and shortcomings of their method (lin 311-313, line 316, line 294, line 399, and lines 350-352). Indeed, how will we identify bacteria when the curated reference database fails and who will work to identify what needs to be added to the database in the future? Given the critical shortage of medical technologists and their utility at multiple steps in the process, the suggestion that this algorithm + WGS eliminates the need for them is offensive to this reader. One might argue that their numbers are decreasing in part because they are not seen as valued members of the clinical team and due to predictions of their replacement by automation which have proven premature

b) In the second point, line 507, virulence and drug resistance can not necessarily be "ascertained immediately" - especially for some antimicrobial resistance phenotypes which derive from multiple de novo SNVs or other changes, not just resistance gene presence / absence.

c) Also, clinicians have difficultly interpreting resistance gene data and how is this addressed? There are publications which cover these topics none of which are cited. In line 511, relatedness can be deceiving in clinical environments. For instance there can be patients admitted around the same time with similarly related bacterial infections acquired elsewhere, and bacteria vary at differing rates depending on species, complicating analysis. These topics should be addressed.

d) In point 4, the statement "Normal ... diagnostics provide no insight into how organisms change over time" is simply not true. Antimicrobial resistance data is followed by microbiologists and infection control, and yearly antibiograms are produced, to track and follow changes in organisms over time and in fact can be tracked in a single patient to see how resistance is acquired over time which is relevant to clinical care.

e) Line 517 would reconsider the use of the word "mating" also there are numerous mentions in the last section of the relevance to public health in a section devoted to the utility of sequencing to diagnostics, in particular.

f) In sum total for this section (499-519), I would urge the authors to reconsider their statements and their impact on readers and fellow healthcare professionals, and temper their enthusiasm for the technique with a dash of their previously acknowledged limitations many of which have been previously described in the literature but are not cited or mentioned in the manuscript.

Author response:

In response to these useful comments, we have rewritten that portion of the discussion as follows:

While the use of whole genome sequencing for diagnostic microbiology may seem unnecessary, its routine use for this purpose is potent in many respects. Traditional diagnostic bacteriology is a trade that can in many cases require enormous skill and experience. While WGS cannot replace this, it can provide a comprehensive means of identification that can be trained more quickly, and performed systematically. Additionally, when genomic sequences are generated in the course of identification, the potential to derive medical and public health benefit from the information from identified genomes becomes realized. This includes the ability to more rapidly assess cases phylogenetically, in furtherance of epidemiology and disease control. It is possible that in the near future, both drug resistance and virulence factors could also be discerned confidently from genomic data, which would potentially eliminate the need for additional steps in the diagnostic workflow.

Reviewer comments:

6) Methods - software needs versions mentioned throughout. I appreciate the thorough documentation and provision of the code online.

Author response:

We have added the version numbers for all software.

Reviewer comments:

7) Figures - Legends are well detailed, the figures in the supplied PDF appear grainy / aliased / pixelated. Otherwise fine.

Author response:

Thank you for this acknowledgement, we will work with the editors to ensure that the most high-quality figures are included for the figures.

Reviewer comments:

8) One final comment, I feel like this is a topic which has been published on multiple times in the past decade and that a variety of literature exists to support the authors discussions but which is not cited in the manuscript. I would encourage the authors to make sure that all claims are either supported by the data presented or the relevant literature is cited.

Author response:

We have cited additional algorithms as we all stand on the shoulders of giants.

---

## [Decision Letter · Decision Letter 1]

27 Oct 2022

PONE-D-22-20500R1GAMBIT (Genomic Approximation Method for Bacterial Identification and Tracking): A methodology to rapidly leverage whole genome sequencing of bacterial isolates for clinical identificationPLOS ONE

Dear Dr. Hess,

Thank you for submitting your manuscript to PLOS ONE. After careful consideration, we feel that it has merit but does not fully meet PLOS ONE’s publication criteria as it currently stands. Therefore, we invite you to submit a revised version of the manuscript that addresses the points raised during the review process. Your revised manuscript has been reviewed and a minor revision is suggested. Please follow the reviewer comments and make all necessary revision.

We look forward to receiving your revised manuscript.

Kind regards,

Chih-Chieh Chen, Ph.D.

Academic Editor

PLOS ONE

Journal Requirements:

Additional Editor Comments:

Your revised manuscript has been reviewed and a minor revision is suggested. Please follow the reviewer comments and make all necessary revision.

Reviewers' comments:

Reviewer's Responses to Questions

**Comments to the Author**

1. If the authors have adequately addressed your comments raised in a previous round of review and you feel that this manuscript is now acceptable for publication, you may indicate that here to bypass the “Comments to the Author” section, enter your conflict of interest statement in the “Confidential to Editor” section, and submit your "Accept" recommendation.

Reviewer #2: (No Response)

Reviewer #3: All comments have been addressed

2. Is the manuscript technically sound, and do the data support the conclusions?

Reviewer #2: Partly

Reviewer #3: Yes

3. Has the statistical analysis been performed appropriately and rigorously? 

Reviewer #2: I Don't Know

Reviewer #3: Yes

4. Have the authors made all data underlying the findings in their manuscript fully available?

Reviewer #2: Yes

Reviewer #3: Yes

5. Is the manuscript presented in an intelligible fashion and written in standard English?

Reviewer #2: Yes

Reviewer #3: Yes

6. Review Comments to the Author

Reviewer #2: I appreciate the authors attention to detail in making the necessary corrections to the manuscript, including my rather passionate defense of clinical microbiology. I would only ask for a few minor changes to that portion of the response.

Line 570 - change "trade" to "profession" or even "work" - although it should not carry a derogatory connotation, trade in this instance seems dismissive.

Line 572 - "more quickly" - i'm not sure this claim is backed up by the data presented and no data is cited here to back up this claim. Given the time involved in WGS sample preparation, sequencing, and bioinformatics analysis compared

to routine lab workflow. Would strike unless this can be supported.

Line 574 - simply doing the sequencing doesn't realize these potential benefits... there is additional downstream work that has to be done post sequencing... so I would qualify as "can be realized"

Reviewer #3: (No Response)

7. PLOS authors have the option to publish the peer review history of their article (what does this mean?). If published, this will include your full peer review and any attached files.

Reviewer #2: No

Reviewer #3: No

---

## [Author Response · Author response to Decision Letter 1]

27 Oct 2022

Response to Reviewers

Below are the changes we made to the latest revised manuscript of GAMBIT (Genomic Approximation Method for Bacterial Identification and Tracking): A methodology to rapidly leverage whole genome sequencing of bacterial isolates for clinical identification”.

Reviewer #2: I appreciate the authors attention to detail in making the necessary corrections to the manuscript, including my rather passionate defense of clinical microbiology. I would only ask for a few minor changes to that portion of the response.

Line 570 - change "trade" to "profession" or even "work" - although it should not carry a derogatory connotation, trade in this instance seems dismissive.

“trade” was changed to “profession”

Line 572 - "more quickly" - i'm not sure this claim is backed up by the data presented and no data is cited here to back up this claim. Given the time involved in WGS sample preparation, sequencing, and bioinformatics analysis compared

to routine lab workflow. Would strike unless this can be supported.

Stuck the works “trained more quickly, and” the sentence now reads

“While WGS cannot replace this, it can provide a comprehensive means of identification that can be performed systematically.”

Line 574 - simply doing the sequencing doesn't realize these potential benefits... there is additional downstream work that has to be done post sequencing... so I would qualify as "can be realized"

Changed “realized” to “can be realized”

---

## [Editor Report · Decision Letter 2]

31 Oct 2022

GAMBIT (Genomic Approximation Method for Bacterial Identification and Tracking): A methodology to rapidly leverage whole genome sequencing of bacterial isolates for clinical identification

PONE-D-22-20500R2

Dear Dr. Hess,

We’re pleased to inform you that your manuscript has been judged scientifically suitable for publication and will be formally accepted for publication once it meets all outstanding technical requirements.

Kind regards,

Chih-Chieh Chen, Ph.D.

Academic Editor

PLOS ONE

---

## [Editor Report · Acceptance letter]

14 Nov 2022

PONE-D-22-20500R2 

GAMBIT (Genomic Approximation Method for Bacterial Identification and Tracking): A methodology to rapidly leverage whole genome sequencing of bacterial isolates for clinical identification 

Dear Dr. Hess:

I'm pleased to inform you that your manuscript has been deemed suitable for publication in PLOS ONE. Congratulations! Your manuscript is now with our production department. 

Kind regards, 

on behalf of

Dr. Chih-Chieh Chen 

Academic Editor

PLOS ONE